# A Supervised Machine-Learning Prediction of Textile's Antimicrobial Capacity Coated with Nanomaterials

Mahsa Mirzaei [1], Irini Furxhi [1,2,*], Finbarr Murphy [1,2] and Martin Mullins [1]

1   Department of Accounting and Finance, Kemmy Business School, University of Limerick, V94 PH93 Limerick, Ireland; mahsa.mirzaei@ul.ie (M.M.); finbarr.murphy@ul.ie (F.M.); martin.mullins@ul.ie (M.M.)
2   Transgero Ltd., Cullinagh, Newcastle West, V42 V384 Limerick, Ireland
*   Correspondence: irini.furxhi@transgero.eu; Tel.: +35-38-5106-9771

**Abstract:** Textile materials, due to their large surface area and moisture retention capacity, allow the growth of microorganisms, causing undesired effects on the textile and on the end-users. The textile industry employs nanomaterials (NMs)/composites and nanofibers to enhance textile features such as water/dirt-repellent, conductivity, antistatic properties, and enhanced antimicrobial properties. As a result, textiles with antimicrobial properties are an area of interest to both manufacturers and researchers. In this study, we present novel regression models that predict the antimicrobial activity of nano-textiles after several washes. Data were compiled following a literature review, and variables related to the final product, such as the experimental conditions of nano-coating (finishing technologies) and the type of fabric, the physicochemical (p-chem) properties of NMs, and exposure variables, were extracted manually. The random forest model successfully predicted the antimicrobial activity with encouraging results of up to 70% coefficient of determination. Attribute importance analysis revealed that the type of NM, shape, and method of application are the primary features affecting the antimicrobial capacity prediction. This tool helps scientists to predict the antimicrobial activity of nano-textiles based on p-chem properties and experimental conditions. In addition, the tool can be a helpful part of a wider framework, such as the prediction of products functionality embedded into a safe by design paradigm, where products' toxicity is minimized, and functionality is maximized.

**Keywords:** nanoparticles; nanomaterials; antimicrobial; antibacterial; textiles; machine learning



## 1. Introduction

Textile materials are susceptible to microbial contamination due to their structural properties in combination with numerous environmental factors such as humidity and temperature [1]. The growth of harmful microorganisms imposes a range of impacts on the textile, including the generation of unpleasant odour [2], stains and discoloration [3], reduction of fabrics' mechanical strength [4], and an increased probability of contamination to the end-user [5]. Therefore, it is important to impart antimicrobial finishes on textile materials to minimize the growth of microbes during their usage [6]. Microbes have the ability to multiply and survive on the textile from days to months, and colonization of the pathogenic microorganisms on surfaces lead to the formation of biofilms that can become inaccessible to antimicrobial agents [7,8]. Antibiotics are used to tackle persistent pathogenic microorganisms; however, their overuse leads to antibiotic-resistant microorganisms and chronic infection diseases [9]. In the healthcare setting, biofilm-associated diseases account for over 80 percent of infections, resulting in increased patient morbidity and significant medical burdens [10]. Therefore, numerous studies and investigations have focused on antimicrobial treatment of textile materials, which entails a set of requirements, namely, (1) efficiency against a broad spectrum of species [11]; (2) no adverse effect on the quality of the textile [12]; (3) durability in the course of laundering, dry cleaning,

and hot pressing; (4) compatibility with textile chemical processes such as dyeing [13]; and (5) biocompatibility, which consists of cytotoxicity, sensitization, and irritation testing before being marketed [14].

### 1.1. Nanotechnology for Functional Textiles

Growing concern around multidrug-resistant microbes has propelled the development of novel, effective, and long-term antimicrobial and biofilm-preventing materials [15]. Nanotechnology and nano-enabled finished materials have received considerable attention recently (https://www.businesswire.com/news/home/20190807005533/en/The-Global-Market-for-Nanotextiles-2019-2030-Revenues-by-Applications-Nanomaterials-Type---ResearchAndMarkets.com (accessed on 10 December 2021)). Nanotechnology involves materials at dimensions of 1 to 100 nm with distinctive properties from their bulk [16]. These nanomaterials (NMs) have gained interest due to their unique properties and extensive range of industrial applications in textiles, cosmetics, food, coatings, pharmaceuticals, medical, sports, aviation and space, batteries, solar hydrogen, construction, optics, thermoelectric devices, and other day-to-day products [17]. NMs improve textile's resistance to microbes, increase their capacity to absorb dyes, and modify their wettability allowing efficient functionalization without altering textile properties [18]. Recent H2020 European Projects, i.e., ASINA (http://www.asina-project.eu/ (accessed on 10 December 2021)) and DIAGONAL (https://diagonalproject.eu/ (accessed on 10 December 2021)) are focusing on the development of sustainable by design nano-enabled products such as textiles that will require data regarding functionality analysis.

Antimicrobial NMs are used in conjunction with antibiotics to inhibit the growth of multidrug-resistant bacteria and to provide the textile a multifunctional modification [19,20]. The antibacterial capabilities of NMs (organic and inorganic) are related to the inability of microorganisms to build resistance to the latter [21]. In the textile industry, many organic and inorganic NMs have been utilized, i.e., inorganic metal and metal oxide NMs (M and MO-NMs) such as silver (Ag) [22], iron oxide ($Fe_3O_4$), titanium oxide ($TiO_2$) [23], zinc oxide (ZnO) [24], copper oxide (CuO) [25], silicon dioxide ($SiO_2$) [26], magnesium oxide (MgO) [27], zirconium oxide ($ZrO_2$) [28], gold (Au) [29], and organics such as chitosan (Ch) [30] or caesium (Cs) [31]. These NMs are widely used as antimicrobial agents in the textile industry and are considered suitable [32] due to their low toxicity to human cells [33], low cost [34], inhibition effect against a broad range of bacteria, and inhibition of biofilm formation [35]. A variety of modes of action are involved in the antimicrobial mechanisms of NMs. The major pathways are the

(1)　production of reactive oxygen species (ROS) and oxygen-free radicals in the microbial cells [36];
(2)　disruption of the microbial cell membrane [37,38];
(3)　alteration of microbial signal transduction pathways [39,40]; and
(4)　release of metal ions and interaction with biomolecules inhibiting their function and consequently the bacterial replication. These pathways are related and can occur simultaneously [41,42].

Similarly, the antimicrobial potential of NMs is influenced by their physicochemical (p-chem) characteristics, such as size, shape, composition, and structure [35]. Additionally, parameters such as the type of bacteria (Gram-positive and Gram-negative), physiological state of the bacteria (planktonic, biofilm, growth rate), and environmental factors contribute to the bacteria susceptibility towards NMs [43].

The antimicrobial capacities of textiles that are treated with NMs (nano-textiles) are affected by many factors such as deposition method and washing conditions [44].

(a)　There are several methods used for deposition of NMs on textiles, depending on the type of NM and fibre. The NMs are either incorporated into the fibres during extrusion or attached to the surface during finishing. Physical methods such as plasma pre-treatment, irradiation, or ultrasound, and chemical methods, for example, chemical reduction in aqueous media, electrochemical reduction, sonochemistry

synthesis, and chemically assisted radiation, are the most widely employed techniques for nano-textile production [45,46]. The finishing methods can be dip-pad technique [47], pad-dry-cure processes [48], spraying [49], foam finishing [50], grafting [51], layer by layer assembly [52], dip coating [53], impregnation process [54], exhaustion method [55], microwave-assisted deposition [56], ultrasonic agitation [57], ultrasound irradiation [58], vapor deposition [59], chemical reduction deposition [60], drop-coating [61], sputtering [62], sol-gel [63], and electroless deposition [64].

(b)   Textiles are exposed to a range of circumstances during their lifespan, including washing, heat, and dry cleaning; in some instances, it is important to know how well the textile can preserve its antimicrobial capacities. Therefore, the effect of washing conditions on the antimicrobial capacity of nano-textiles can be determined by (1) various durability tests (laboratory scale, industrial and domestic washing machines), (2) type of detergent where applicable, (3) amount of water used (tap or distilled), (4) number of washing cycles, and (5) temperature [65].

The determination and evaluation of the antimicrobial capacities of the nano-textiles are accomplished by several methodologies, with the following two categories being the most widely used: the agar diffusion test (qualitative or semi-quantitative evaluation) and dynamic shake flask test (quantitative evaluation) [66,67]. The assessment can be influenced by growth conditions, bacterial density, test duration, and type of material.

### 1.2. Machine Learning

Machine learning (ML) is a subset of artificial intelligence (AI) that covers a wide variety of modelling tools used for a vast range of data processing tasks; it has gained popularity over the last decade across the majority of scientific disciplines [68]. There are many studies that have employed ML models to primarily forecast NM's safety and toxicity [69–71]. Mirzaei, et al. [72] developed a tool to predict the antimicrobial capacity expressed as zone of inhibition of various NMs using regression models. The authors used data from in vitro experimental set-ups. In this work, we present a ML model to predict the antimicrobial efficiency and durability of nano-textiles after several washes. This tool predicts the antimicrobial capacity by investigating the p-chem properties of NMs, exposure conditions, and the bacteria type as inputs. Building on Mirzaei, et al. [72], in this work, we predicted the antimicrobial capacity expressed as a percentage (microbial viability). In addition, we expanded the applicability domain of the type of NMs by also including organic NMs. This paper focused on nano-enabled products (nano-textiles), whereas in the previous paper, the attention was focused on NMs exposed in vitro. Notably, in this work, we considered the nano-textile's durability by integrating information regarding the antimicrobial capacity after laundering and the washing cycles. A significant key difference in this work was the inclusion of experimental conditions as input variables such as the technological application method, type of substrate, washing method, washing cycles, and the techniques to evaluate the antimicrobial efficacy. Our objective was to develop a ML tool that uses information related to NM's properties and exposure conditions against a range of organisms (bacteria and fungi) as inputs and growth inhibition measurements as outputs to predict nano-textiles with greater antimicrobial capacities. This tool assists researchers in quickly identifying nano-textiles with effective antimicrobial capabilities and thereby prevents the spread and proliferation of harmful microorganisms. This method is cost- and time-effective as it reduces the number of tests required and thus enhances efficiency in laboratories.

## 2. Materials and Methods

### 2.1. Approach

Figure 1 illustrates the model's implementation roadmap. Studies on the antimicrobial effects of nano-enabled textiles were collected and data extraction was performed manually for NMs' p-chem properties, technological application method, exposure conditions, washing cycles, methodology, and the type of exposed bacteria. The initial dataset (Dataset I)

was evaluated for completeness and variables with high missing values were discarded. Data pre-processing, including harmonization, standardization [73], one hot encoding, and data splitting [74], was performed to increase predictability. Several regression models were trained and validated against a range of performance metrics. Finally, attribute importance analysis was carried out to identify the variables that most influence the prediction of results [75].

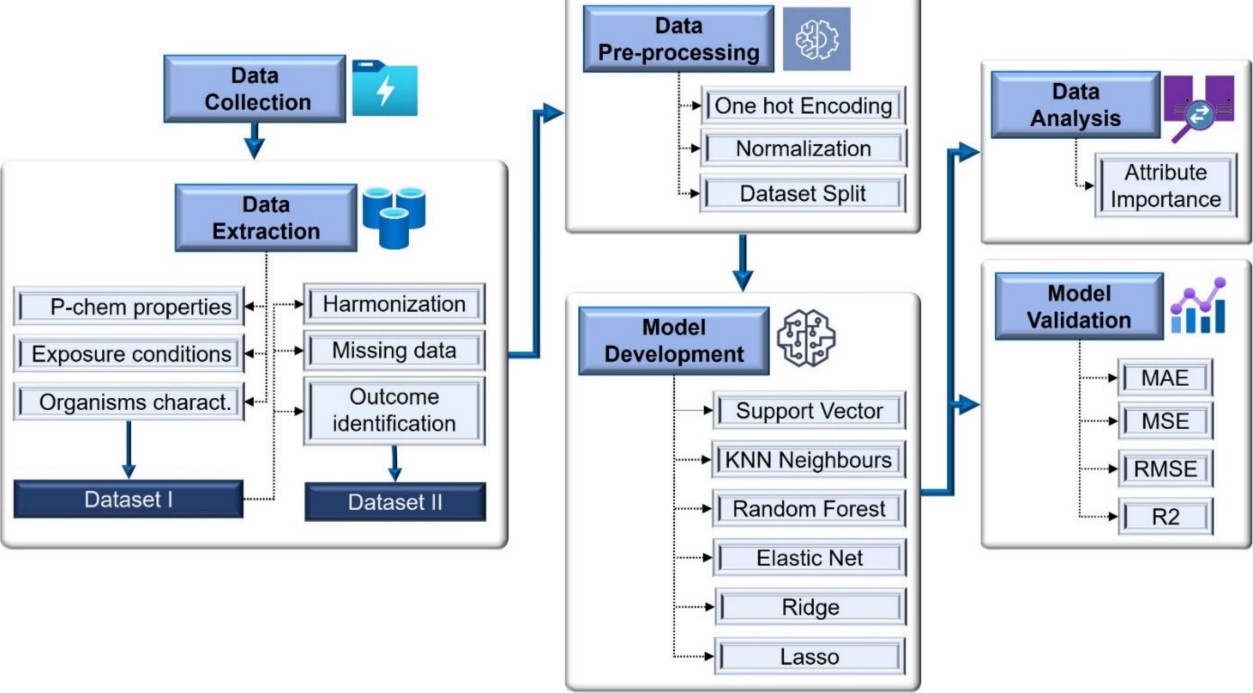

**Figure 1.** Model development workflow. We employed different regression models. Evaluated by mean absolute error (MAE), mean square error (MSE), root mean square error (RMSE), and coefficient of determination or R-squared (R2).

### 2.2. Data Collection

A literature search was conducted on articles published between 2010 and 2021, and 129 articles that studied the effect of nano-enabled textiles to eliminate or inhibit microbial growth after washes or without washing were selected. The search included various keywords such as "antibacterial", "antimicrobial", "nanoparticles", "nanomaterials", "textile", "washes", and "bactericidal" effects and/or assessment. Inclusion requirements consisted of studies in the English language and original in vitro studies with numerical outcomes. The rationale for the exclusion criteria included reviews, case reports, studies with binary results, and studies that demonstrated results in figures.

### 2.3. Data Extraction

*Input extraction*: Each paper was reviewed with a focus on (1) the nano-specific descriptors such as the type, core size, and shape [17,76]; (2) the experimental parameters of the study design (use of a binder, fabric/substrate, number of washes, application technology, type of durability test, and detergent [77]); and (3) exposure conditions, i.e., dose, duration, and bacteria exposed. The aforementioned variables were taken as input (feature) attributes for the prediction of the antimicrobial durability of the investigated nano-textiles. We gathered several studies on inorganic and organic NMs (Ag, $Fe_3O_4$, $TiO_2$, $SiO_2$ MgO, CuO, ZnO, $ZrO_2$, Au, Ch, Cs) since they are widely used as antimicrobial agents in the textile industry [32,78].

*Outcome extraction*: There are various analyses and techniques to assess antimicrobial efficacy. In this study, two main antimicrobial measures were reported as outcomes to train our model. On the basis of the reviewed studies, we found that 27% reported zone

of inhibition (ZOI) and 73% reported the percentage of bacterial reduction before and after washing cycles (where applicable). Studies not performing washing cycles to test the antimicrobial durability but executing application methods to coat the substrate were also included. In the latter case, the washing cycles were set to zero.

### 2.4. Data Pre-Processing

Following the selection of the outcome and the assessment of data completeness in dataset I, we created the final dataset (dataset II), 1676 rows and 15 columns with few missing values among the inputs. Regression models underperform with null data; therefore, missing values were replaced with the mean value generated for each column. Managing the missing value maintains dataset consistency and helps it to be more coherent and unified [69].

In regression models, categorical variables are converted into integers/numerical dummy variables. One hot encoding was performed, wherein each category value was transformed into a new column, and a value of 1 or 0 (true/false) was allocated to each column [74]. Numeric variables measured at different scales can cause bias and do not contribute equally to model fit; therefore, numerical input values were normalized to enhance the model performance [79]. Several normalization methods, such as min–max, Log10, and mean (average) were used for a more precise evaluation with calculating skewness. If skewness is between $-/+1$ and $-/+\frac{1}{2}$, the distribution is moderately skewed. A supervised algorithm to predict outputs of an unknown target function, it is necessary a training set to be provided prior to model execution. The dataset was randomly split into two subsets, one to train the model (training set) containing 80% of the dataset for measuring robustness and the rest (20%) for measuring the predictability of the model (test set).

### 2.5. Regression Models

In this study, the regression model found the correlation between NMs' p-chem properties and experimental conditions to the percentage of bacteria reduction, enabling the prediction of the antimicrobial durability of nano-textile after several washes. Diverse regression algorithms were employed as potential candidates to explore which model delivered the most accurate prediction. The least absolute shrinkage and selection operator (LASSO) regression, ridge regression (RR), elastic net regression (ENR), random forest (RF), *k*-nearest neighbours (KNN), and support vector regression (SVR) were explored in this study. Models were built in Python version 3.7.6, Scikit-learn version 0.24.1.

LASSO regression is a popular variable selection and shrinkage estimation method [80]. It functions by imposing a restriction on input variables, which then "shrinks" and minimizes the regression coefficients close to zero, resulting in a more relevant set of predictors in order to reduce overfitting [81].

RR is a simple approach that works almost in the same way as the LASSO. RR model provides different importance weights to the features but does not drop unimportant features in comparison with LASSO [82].

ENR utilizes the penalties from both the LASSO and RR models to regularize regression models [83]. ENR often outperforms LASSO, specifically when the number of predictors is larger than the number of observations [83]. This method can improve predictions by performing variable selection by forcing the coefficients of "non-significant" variables towards zero [81].

RF comprises of various decision trees that are trained independently on a random subset of observations. RF can work well with large datasets and manage numerous variables without altering the accuracy and handle the missing values while preventing overfitting [84]. As a classifier, RF performs feature selection, using a small subset of "strong variables" for the classification resulting in superior performance (with higher accuracy in prediction) [85]

SVR learns by assigning labels to objects, and it is employed in cases of both classification and regression [86]. In the SVR, each data item is conceived as a point in *n*-dimensional space with the value of each input related to the value of a specific coordinate.

KNN is a lazy learning algorithm that utilizes all the data for training whilst categorizing a new data point or instance. It works by calculating the Euclidean or Manhattan distance between a new data point to all other training data points. It selects the *k*-nearest data points, where K can be any integer assigning the data point to the class to which the majority of the K data points belong [87].

### 2.6. Model Validation

The main objective of this paper was to develop a model that has a high predictive capacity. *k*-fold cross-validation was employed to prevent overfitting and ensure robustness [88,89]. During cross-validation, the model was trained using parts of the training set (i.e., 70% or 80%) and leaving a subset for subsequent testing (i.e., 30% or 20%) [90,91]. In regression models, adjusting hyperparameters helps to produce an optimal model where both underfitting and overfitting are minimized. In LASSO, RR, and ENR models, different sets of alpha values were used to evaluate the models (data not shown). A grid search approach was performed to tune LASSO, RR, and ENR models; this method optimizes the model by keeping the most important features. Finally, the models were evaluated by mean absolute error (MAE), mean square error (MSE), root mean square error (RMSE), and coefficient of determination or R-squared ($R^2$). The $R^2$ or the coefficient of determination is a statistical measurement of the proximity or closeness of the real data to the prediction (or fitted regression line).

### 2.7. Important Attribute Analysis

Attribute importance ranks the input features that have the greatest influence on the prediction [92]. Attribute importance is produced by the RF model and is based on Gini's importance, an all-nodes accumulating quantity that indicates how often a particular attribute was selected for a split, and how large its overall discriminative impact was in the regression [84]. Information values range from 0 to 1, with 1 indicating maximum information gain.

## 3. Results

We selected 129 studies that investigated the antimicrobial capacities of nano-textiles before and after several washes (percentage of bacterial reduction) were selected. A total of 110 studies were deemed relevant to our study. The final dataset comprised 1676 rows and 15 columns (14 inputs and 1 output).

### 3.1. Data Pre-Processing

The input variables were selected on the basis of (1) the NM's p-chem properties, (2) experimental design, and (3) exposure conditions. The numerically reported variables are core size, duration, concentration of NMs used for coating the substrate, and the outcome; the rest of the variables are nominal (Table 1).

The missing values of the input variables are size (4.3%), shape (11.7%), concentration (18.5%), durability test (12.5%), and detergent (28.8%). The detergent column was removed due to numerous missing values. The missing values in size, concentration, and duration columns were replaced by the mean of the values in each column. In this study, the experimental methods (zone of inhibition or percentage reduction) for the assessment of antimicrobial properties of textiles were extracted as output variables. Due to a large number of missing values of the zone of inhibition (74%, data not shown), we retained the percentage reduction (of the microorganisms in the textile) as the outcome.

**Table 1.** The preliminary and final input variables in dataset I and dataset II.

| | | | Dataset I | | Dataset II |
|---|---|---|---|---|---|
| | **Variables** | **Type** | **Min-Max, Mean, or Label** | **Data Transformation** | **Min-Max, Mean, or Label** |
| **P-chem properties of NM** | Primary size | Numeric | 0.65–500, 42.95 (nm), NaN | Selected, normalized | 0.187–2.69, 1.45 |
| | NM type | Nominal | CuO, Ag, ZnO, Au, Ce-ZnO, $ZrO_2$, $Fe_3O_4$, Mn, Co, CuO-$TiO_2$, $TiO_2$, SA-TSA, ZnO-Cs, Cs, $SiO_2$-Ag-Cu, Ce, $Fe_3O_4$-ZnO | Selected and simplified | Ag, Au, Ce, Ce-ZnO, CS, CuO, CuO-$TiO_2$, $SiO_2$-Ag-Cu, $TiO_2$, ZnO, others |
| | Shape | | Ellipsoidal, spherical, crystalline, rod, wire, irregular, rectangular, hexagonal, others | | Hexagonal, spherical, rod, others |
| | Binder | Binary | Yes or no | Selected | Yes or no |
| **Exposure conditions Experimental study design** | Concentration | Numeric | 0–33.25, 2.92 (µg/mL), NaN | Selected, Normalized | −4.6–1.5, 2.9 |
| | Duration | | 0–52, 22.79 (h), NaN | | −3.6–4.62, 2.86 |
| | Substrate | Nominal | Cotton, polyethylene terephthalate (PET), viscose, cotton-polyester, polyamide, polyester, wool, silk, wool polyester, bamboo, denim | Selected and simplified | Bamboo, polyester, cotton, others |
| | Washing cycles | Numeric | 0–50, 9, NaN | Selected | 0–50, 9 |
| | Durability test | Nominal | Industrial, domestic, and commercial washing machines; agitation; boiling; bath; NaN | Selected and simplified | Agitation, domestic, domestic and commercial, industrial, others |
| | Detergent | | Nonionic, standard, commercial, water, anionic, commercial, NaN | Eliminated due to high NaN | - |
| | Application method | | Sonochemical, dip coating, exhaust, immersion, grafting, sorption, padding, spraying, blade coating | Selected and simplified | Dip coating, immersion, padding, sonochemical, others |
| | Evaluation standard | | ISO_20743, AATCC_100, AATCC_147, GB_T_20944_AATCC_61, AATCC_147_ISO_20645, ISO_20645, ASTME_2149, AATCC_30, ASTM_2180 | Eliminated | - |
| | Evaluation method | | Agar diffusion, dynamic shake flask | Selected | Agar diffusion, dynamic shake flask |
| | Washing temperature | Numeric | 20–95, 40, NaN | | - |
| | Method of synthesis of NM | Nominal | Biosynthesis, degradation, dip-coated temp-curated ultrasound, ex situ synthesis, in situ biosynthesis, in situ deposition (alkalization and deposition), in situ microwave irradiation, in situ reductions, in situ sol–gel immersion, in situ ultrasound irradiation, ionic gelation, photochemical reduction, reduction of cellulose in viscose, reverse micellar cores, sol–gel, sonochemical, ultrasound irradiation, wet chemical method | Eliminated due to high NaN | - |
| **Bacteria** | Organisms | Nominal | *Acinetobacter baumannii, Alternaria brassicicola, Aspergillus niger, Bacillus Subtilis, Candida albicans, Escherichia coli, Enterococcus Faecalis, Fusarium oxysporum, Klebsiella aerogenes, Klebsiella pneumoniae, Microsporum canis, Methicillin-resistant Staphylococcus aureus, Pseudomonas aeruginosa, Staphylococcus aureus, Staphylococcus epidermidis, Streptococcus pyogenes, Salmonella typhimurium, Trichophyton mentagrophytes* | Selected and simplified | Gram-negative, Gram-positive, fungus |

The organism variable (bacteria and fungi) was dispersed across 18 different types (Figure 2A). To avoid overfitting the model, we merged them into three categories: Gram-positive, Gram-negative, and fungi. In the case of NM deposition in textile, 10 different techniques (application methods) were reported, which we merged into 5 groups (Figure 2B). The same was applied for the shape, substrates, and washing durability test variables that were simplified and grouped in dataset II (Table 1).

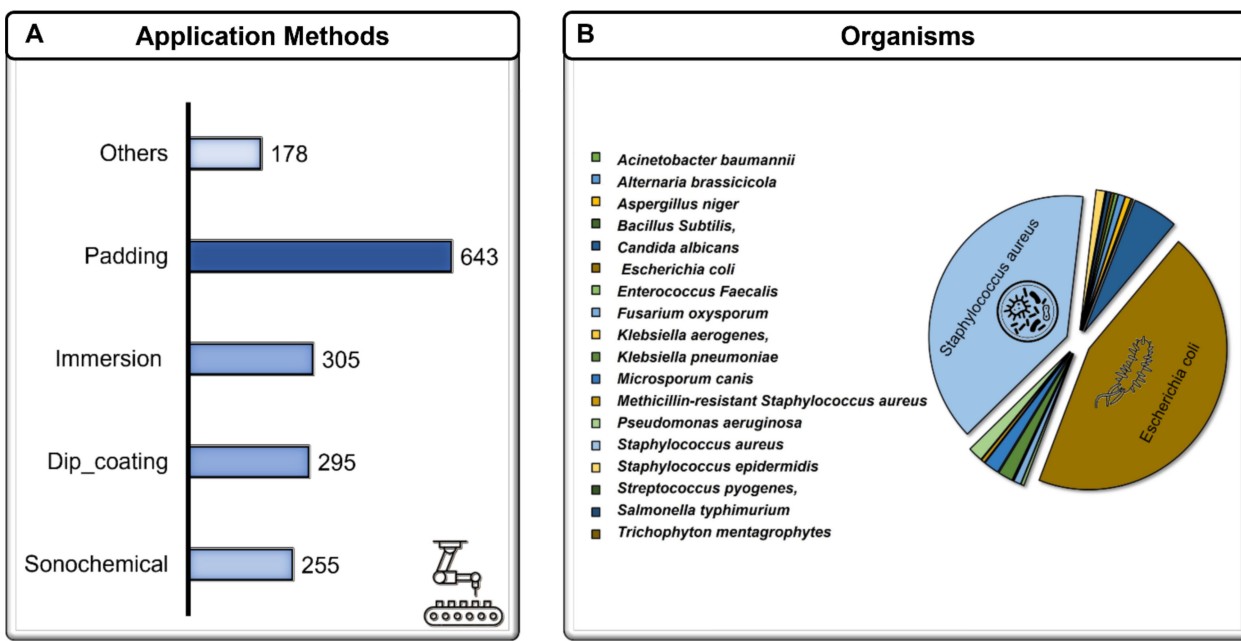

**Figure 2.** (**A**) Different application methods; (**B**) different investigated organisms (bacteria and fungus) used as input.

Numerical values were normalized by three techniques: log10, mean (average) and min-max (Figure 3) and evaluated based on skewness. Normalization by mean had skewness closer to zero for the duration input. Log10 skewness was closer to zero for the primary size and concentration variable.

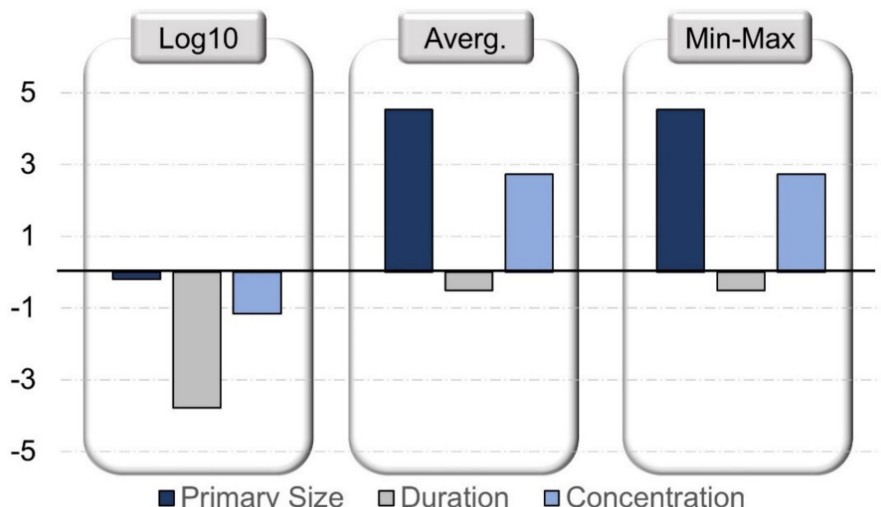

**Figure 3.** Data skewness of numerical inputs (primary size, duration, and concentration) was based on the three different normalization techniques (log10, mean, and min−max).

### 3.1.1. Validation of the Models

On the basis of the evaluation and validations methods, we found that RF outperformed the other models (LASSO, RR, ENR, SVR, and KNN) (Figure 4). The RF model

demonstrated the lowest error compared to the other models with RMSE, MAE, and RMSE at 14.8, 8.3, and 221.3, respectively, and the highest $R^2$ score at 0.7.

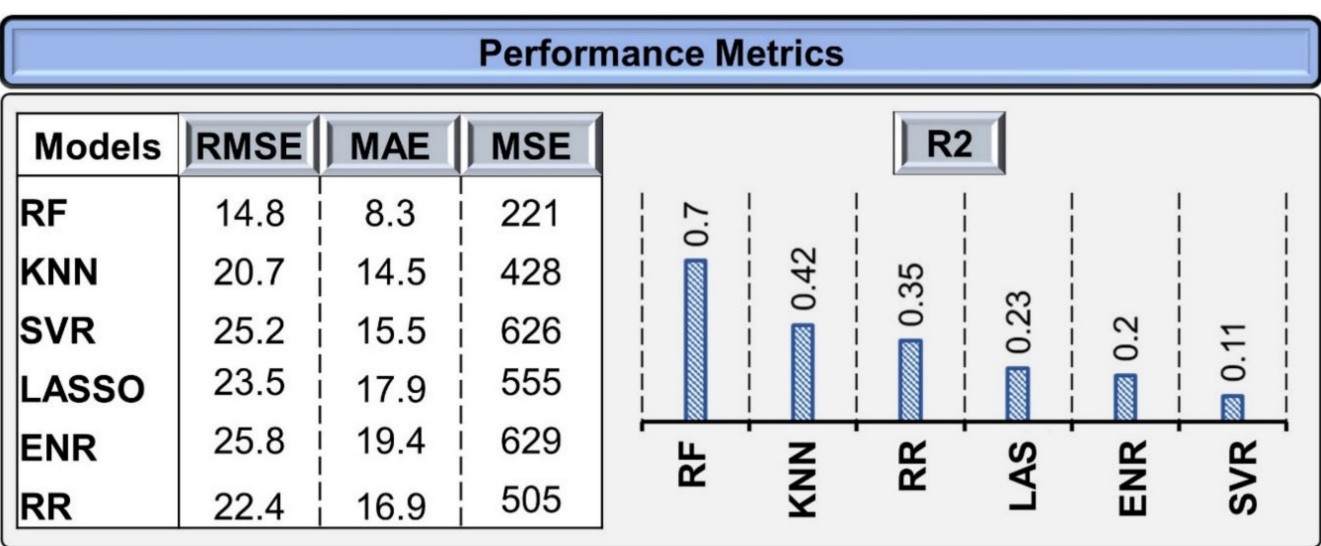

**Figure 4.** Performance metrics of the models employed.

### 3.1.2. Important Attribute Analysis

Figure 5 presents the results of attribute importance analysis. The type of NM is the most important attribute that determines the efficacy of antimicrobial nano-textiles. The application (deposition) method, shape, and durability test were identified as relatively important factors, followed by the NM size, type of substrate, evaluation method, and washing cycles. NM concentration, time, type of binder, and organism exposed had relatively less impact on the prediction of the antimicrobial capacity of nano-textiles. Further data analysis, such as variable correlation, is presented in Supplementary Materials Figure S1.

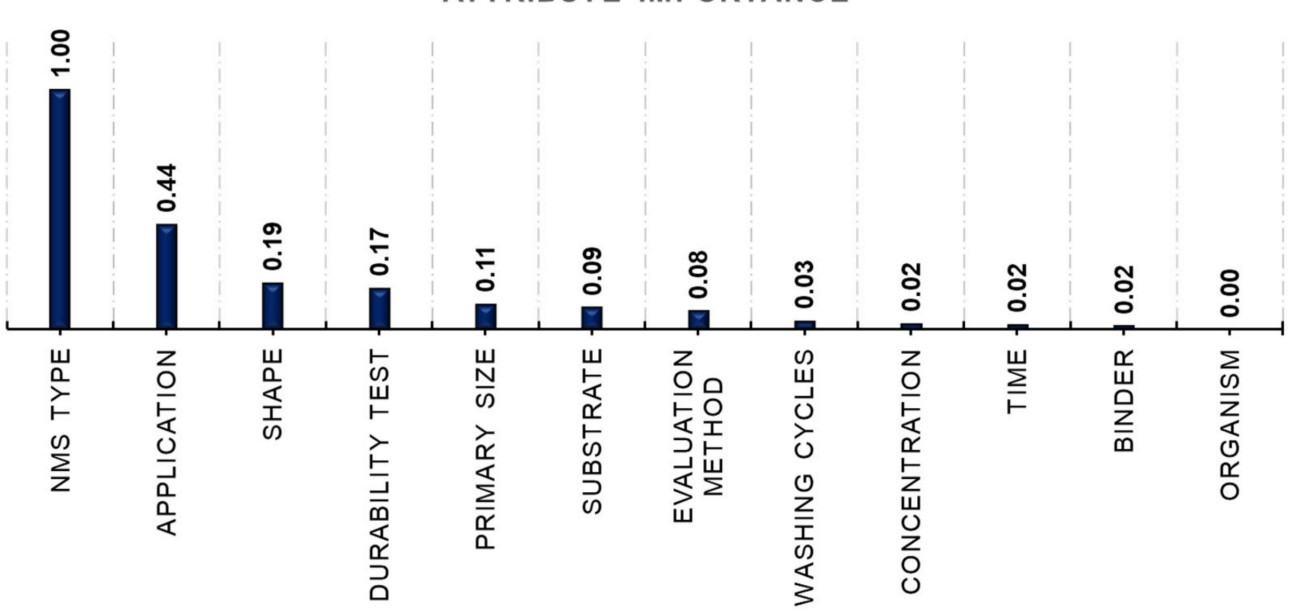

**Figure 5.** Attribute importance analysis by RF model.

## 4. Discussion

In this study, we present a ML tool that predicts nano-textile antimicrobial capacity before and after several washes (up to 50 washes). The objective is to identify the most significant variables that influence the prediction. For this purpose, we manually created a dataset that adhered to the FAIR principles [93], as well as a OECD-compliant model with an explicit algorithm and a well-defined numerical endpoint [94]. The dataset complies with the FAIR principles, since it is findable (https://github.com/mahsa-mirzaei/AMNT.git, 10 December 2021), accessible (free access), interoperable (the descriptors are transparent in their meaning with measurement metrics, and, where feasible, the method of analysis is included), and re-usable.

To analyse our data and determine the impact of each variable on the prediction of the antimicrobial efficacy of the nano-textile, we employed different regression techniques, namely, LASSO, RR, ENR, SVR, RF, and KNN algorithms. To assess the predictability and accuracy of the model, we used statical approaches such as MSE, MAE, RMSE, and $R^2$. The smaller the values of MSE, RMSE, and MAE, and the closer the $R^2$ value was to 1, the better the model performed, indicating that it is well fitted. According to our models' evaluations result, RF exceeded other algorithms with the highest $R^2$ value (0.7), predicting the antimicrobial efficacy of the nano-textile more precisely. The predictive performance of RF is due to its ability to operate with large number of variables while requiring less data pre-processing. The KNN algorithm was the second-best model for prediction outcome, with an $R_2$ value of 0.4. The LASSO algorithm encourages the coefficient of correlated variables to reach zero, shrinking the parameters and preventing multicollinearity. However, this can be challenging because it results in loss of critical information that might alter the prediction and result in poor accuracy. The RR forces the coefficient close to zero. LASSO with $R^2$ value of (0.23) was quite similar to RR, with an $R^2$ value of (0.35), but with better predictive accuracy due to feature selection properties. The ENR with $R^2$ value of (0.20) merged the strengths of LASSO and RR, applying a mix of λ1 (LASSO-type) and λ2 (ridge-type) penalization. SVR with $R^2$ value of 0.11 requires features scaling and accurate parameters selection. The disadvantages of this model are overfitting and low predictability accuracy.

Since the last decade, ML has been extensively employed in the field of nanoscience. The application of a QSAR perturbation model to predict the antibacterial activity of NM was demonstrated by Speck-Planche, et al. [95]. They used NMs' diverse p-chem characteristics and various experimental settings as inputs for the model training. The outcomes included several endpoints such as minimum bactericidal concentration (MBC), minimum inhibitory concentration (MIC), and microbicidal effect (Microb-Eff). The prediction efficacy was shown to be affected by changes in p-chem characteristics (composition, size, shape) NMs as well as from the experimental conditions (type of bacteria, time, coating). Speck-Planche merged the outcomes into a binary format (active/inactive), whereas in our analysis, we allowed for the prediction of the numerical continuous outcome. They found that the type of bacteria, time, and coating determined the accuracy of the outcome prediction; however, in our study, the type of microorganism, time, and binder/coating agents had no relatively significant effects according to the attribute importance evaluation.

In a more recent paper, Daly, et al. [96] in 2015 demonstrated the use of bagged artificial neural network (ANN) model to discover the link between NM's p-chem characteristics, experimental conditions, and bacterial viability. The study's dataset solely contained Gram-negative bacteria. In comparison to the above paper, we evaluated the antimicrobial efficacy of NMs against Gram-positive bacteria, Gram-negative bacteria, and fungi. In concordance with the results of the two aforementioned studies, we found that the p-chem characteristics were the main factors that affect the accuracy of the outcome prediction. It is noteworthy that none of the above papers included technological variables (method of deposition) or variables related to products in their analysis (nano-enabled products such as the type of fabric). Finally, our analysis incorporated washing durability in the dataset, in addition to the antimicrobial capacity of NMs.

In this study, we reviewed several inorganic and organic NMs because they are widely used as antimicrobial agents in the textile industry [32,78]. According to the literature, numerous factors influence the antimicrobial efficacy and durability of nano-textiles. The most important ones are the application (deposition) method, the p-chem properties of the NM (shape, surface area, and zeta potential), type of NM, type of substrate, and the type of durability test used [97,98].

The attribute importance assessment by the RF model indicated that among the p-chem properties, the type of NM (1.0), shape (0.19), and size (0.1) are the top three significant properties that influence the antimicrobial prediction. Our results are in agreement with the landscape presented in the literature, for example, Inam, et al. [99] indicated that the type, shape, and size of NM are important factors to determine the antimicrobial activity of the NMs. Other studies report similar results [100,101]. Patra, et al. [102], Wu, et al. [103] demonstrated that NMs with smaller sizes have a significant impact on the antimicrobial durability of nano-textiles. Zille, et al. [104] stated that the antimicrobial efficiency of NMs larger than 30 nm was not affected by the concentration of the NMs but was primarily affected by the release of NMs over time. Similarly, Panáček, et al. [105] showed that the antimicrobial efficacy of NMs with different sizes was not dependent on concentration. Our model's result confirmed that the concentration was not as important (0.023) as the p-chem properties on influencing the antimicrobial prediction.

The application/deposition technique (0.439) was the second most important parameter in predicting the efficacy of nano-textiles. We investigated some of the most common chemical (exhaustion, pad-dry-cure, dip-coating) and physical (ultrasound-assisted deposition) methods used for NM deposition in textiles. On the basis of the method of deposition, the antimicrobial nano-textile can act either by contact or diffusion [61,106]. According to the literature, the physical methods are effective for the deposition and synthesis of NMs on textiles, since in chemical deposition, the strong bonding of NMs to the fibres may decrease the antimicrobial effectiveness [107].

Strong adhesion is required between NMs and substrates to maintain and conserve the antimicrobial properties of nano-textiles. The durability of the nano-textile depends on the type of durability test (mechanical stress induced by industrial or domestic washing machines) [98,108]. The durability test employed (0.174) was another important variable found in our study. Some articles investigated the influence of temperature on the durability of the antimicrobial nano-textile and demonstrated that temperature had a great influence. In the studies that we investigated, only 21% of studies reported the temperature. In order for temperature and other external factors influencing the antimicrobial properties of nano-textiles to be studied, it is necessary to have a homogeneous and complete dataset. We highly recommend that future research report as many variables as possible in a systematic and comprehensive manner.

The substrate also had an impact (0.093) on the antimicrobial efficacy of nano-textiles. Rivero, et al. [76] showed that the durability and efficacy of nano-textiles depend on the type of fabric. Cotton is a fabric used for various applications, including medical textiles, apparel, commercial textiles, automotive application, and other similar applications. Natural fabrics (cotton, wool, silk, flax, jute) are more susceptible to microbial attack than synthetic fibres (polyester, cellulose, fiberglass) [109,110].

Studies show that the presence or absence of the binder used can affect the binding of the NM to the substrate or leaching in various environmental compartments [76,111]. In our dataset, to prevent overfitting the models, we reported the binder in a binary format of presence or absence of the binder. The usage of binders had less influence (0.015) in comparison to the other variables on the antimicrobial prediction. This implies that the information on binders is still required in order for RF to successfully predict antimicrobial efficacy.

The reviewed studies captured the antimicrobial efficacy using two different techniques, the agar diffusion (including the AATCC 147, EN ISO 20645, ASTM 2180, BS EN ISO 20743, JSA JIS L 1902, and CEN/TC 248 WG 13 protocols) and the dynamic shake

flask tests (AATCC 100, ASTM E 2149, EN ISO 20743, JIS L 1902 protocols). The evaluation method's influence on the antimicrobial prediction (0.078) is justified [112,113].

The number of washing cycles is relatively important (0.034). Prior studies have noted the importance of the washing cycle on the antimicrobial efficacy of nano-textiles [97,114,115]. The duration of the test was found to be less effective (0.02), and it refers to the time that the nano-textile is in contact with the microorganisms.

The microorganisms have a relative less significant effect on antimicrobial efficacy prediction. Microorganisms adhere differently to fabrics—for example, Staphylococcus aureus adheres strongly to cotton and polyester in comparison to Escherichia coli; however, it has a minor influence on the antimicrobial properties of the nano-textile [116,117]. In other words, depending on the p-chem properties of NM and the type of fabric, the bacteria have different responses. Most bacteria can be grouped into two classes, depending on the structure of their cell wall: Gram-positive and -negative, responding differently to NMs [118]. Our results suggest that p-chem properties have a stronger influence on the prediction than the information of microorganisms per se. For example, Gram-negative bacteria are surrounded by LPS molecules with negative charges. These molecules have a greater affinity towards the positive ions that are released by NMs, which results in increased uptake of ions and intracellular damage [119]. Consequently, it is crucial to report the NM surface properties; this information was missing from the reviewed studies (zeta potential).

This paper introduces a novel usage of ML towards the prediction of antimicrobial capacities of nano-textiles by considering diverse deposition techniques. We bridged various p-chem properties, experimental conditions, and testing approaches to predict the antimicrobial capacity after durability testing. Additionally, we demonstrated the attributes importance in predicting the textile capacity to retain high antimicrobial capacity after washing cycles. It is important to mention that in the selected studies, few studies reported p-chem and surface properties of NM. We emphasize the necessity and importance of reporting the p-chem characteristics such as zeta potential, surface area, and shape of NM because they are crucial in determining the textile efficacy. A harmonized and standardized framework on how to report NMs characteristics and experimental conditions, as well as making those measurements more comparable, is required to improve the forthcoming reporting data. Furxhi, et al. [120] exhibited how to construct and utilize a template for capturing antimicrobial capacity data of NMs or nano-enabled products, promoting the principles of making data scientifically findable, accessible, interoperable, and reusable (FAIR), while encouraging scientists to reuse it. Researchers are expected to follow homogenized frameworks and data logging templates when reporting experimental attributes and outcomes in order to capture data in a format that will allow their re-usability, while increasing data quality. Due to the inconsistency in reporting findings, we have a significant data gap that limits the appropriate application of computational tools. This study underlines the need for a more comprehensive dataset in this field.

## 5. Conclusions

Antimicrobial nano-textiles have attracted the interest of both manufacturers and researchers. We developed a ML tool to predict the antimicrobial efficacy of nano-textiles after several washes. We examined various regression models and discovered that the RF model provides a superior prediction with a 70% accuracy. The attribute importance indicates the significant role of the p-chem characteristics of NM, as well as the application/deposition method, in prediction of the antimicrobial efficacy of nano-textiles. We highlight the significance of standardized and consistent measurements and a harmonized reporting system for p-chem properties and experimental conditions. We shed light on the lack of models related to NM functionality in the literature that might hinder the ongoing effort of safe-by-design frameworks to integrate safety with performance. We raise awareness to the scientific community of the absence of comprehensive datasets concerning the antimicrobial capacity of NMs. Nonetheless, our findings provide a significant contribution

to the field and illustrate the importance of ML-based development tools. It emphasizes the benefits of bringing multiple skill sets to bear on this set of challenges. This approach can assist scientists in predicting the characteristics that impact the antimicrobial capacities of a nano-textile, allowing them to improve nano-textiles and therefore minimizing the growth of harmful microorganisms. The ability to forecast antimicrobial properties of nano-textiles, particularly in healthcare settings, offers enormous potential in terms of providing tangible healthcare benefits.

**Supplementary Materials:** The following are available online at https://www.mdpi.com/article/10.3390/coatings11121532/s1, Figure S1: Seaborn pair-plot correlation analysis of input variables with the outcome.

**Author Contributions:** Conceptualization, M.M. (Mahsa Mirzaei) and I.F.; methodology, M.M. (Mahsa Mirzaei); formal analysis, M.M. (Mahsa Mirzaei); investigation, M.M. (Mahsa Mirzaei); data curation, M.M. (Mahsa Mirzaei); writing—original draft preparation M.M. (Mahsa Mirzaei); writing—review and editing, M.M. (Mahsa Mirzaei), F.M. and M.M. (Martin Mullins); visualization, M.M. (Mahsa Mirzaei) and I.F.; supervision, I.F., F.M. and M.M. (Martin Mullins); funding acquisition, F.M. and I.F. All authors have read and agreed to the published version of the manuscript.

**Funding:** This research was funded by the European Union's Horizon 2020 research and innovation programme under grant agreement no. 862444.

**Institutional Review Board Statement:** Not applicable.

**Informed Consent Statement:** Not applicable.

**Data Availability Statement:** Data available in the Supplementary Material.

**Conflicts of Interest:** No potential conflict of interest was reported by the authors.

## Abbreviations

| | |
|---|---|
| NMs | Nanomaterials |
| P-chem | Physicochemical |
| MO | Metal oxide |
| M | Metal |
| ROS | Reactive oxygen species |
| ML | Machine learning |
| AI | Artificial intelligence |
| MAE | Mean absolute error |
| MSE | Mean square error |
| RMSE | Root mean square error |
| $R^2$ | R-squared |
| ZOI | Zone of inhibition |
| LASSO | Least absolute shrinkage and selection operator |
| RR | Ridge regression |
| ENR | Elastic net regression |
| RF | Random forest |
| KNN | *k*-nearest neighbours |
| SVR | Supervised vector regression |
| nm | Nanometer |
| µg | Microgram |
| ml | Mililiter |
| PET | Polyethylene terephthalate |
| MIC | Minimum inhibitory concentration |
| Microb-Eff | Microb effect |
| MBC | Minimum bactericidal concentration |

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
