# Peer review of "A Supervised Machine-Learning Prediction of Textile’s Antimicrobial Capacity Coated with Nanomaterials"

_coatings, doi:10.3390/coatings11121532_

Round 1
Reviewer 1 Report
The manuscript needs some work before it can be published. Some missing data should be presented in the manuscript. Below are some questions/ comments for the authors to improve the manuscript.
1- Citation section in the first page for “polymers” journal not for “coating”, please revise.
2- The abstract should be concerned with actual results rather than posted information
3- The abbreviation should be mentioned completely the first time, please check throughout the manuscript.
4- Keywords should contain words about the main hypothesis of the study, the author mentioned "Nanoparticles; Nanomaterials" it is the same meaning; " Antimicrobial; Antibacterial" also the same meaning?
5- The introduction contains huge information not related together, and in most cases, the information is repeated. Also, the introduction contains more sub-titles, and this is not preferred in the introduction. therefore authors should rephrase the introduction.
6- The authors should be adding a clear hypothesis at the end of the introduction.
7- Line 74, “(Fe3O4)” should be corrected to “Fe3O4”
8- Line 86, “gram positive and gram negative” should be corrected to “Gram-positive and Gram-negative”.
9- Figure 1 repeated twice, Also, authors should be adding a clear title for figures containing the definition for all abbreviations in figures
10- At the “2.2. Data collection” section, authors said that “A literature search was conducted on articles published between 2010 and 2021 that studied…..”, by simple online search found some published studies not included in the survey such as :
https://doi.org/10.1016/j.carbpol.2012.08.064,
http://doi:10.3390/antibiotics9100641,
https://doi.org/10.1021/acs.iecr.0c04880,
Please check.
11- Materials and Methods contain sentences whose correct place is introduction such as lines 167-172.
12- Line 279, “gram-positive, gram-negative and fungi” should be “Gram-positive, Gram-negative, and fungi”.
13- Line 282, “…test)” delete “)”.
14- Figure 2b, please add the complete scientific name of organisms in figure footnotes
15- Please check the figure numbers throughout the manuscript in the figure title and text.
16- Figure 2, repeated twice on pages 10 and 11, Also, the number of this figure is incorrect.
17- What is the meaning of “Error! Reference source not found” it is repeated more than one time?
18- The manuscript contains numerous sentences that are not clear and require rephrasing.
19- In conclusion, the authors should explain how researchers benefit from this model and its applicability?
20- The manuscript contains a high number of references “135 references”, I recommended reducing it.
21- The manuscript should be subjected to deep English editing.
Author Response
We are grateful for the comments and suggestions by the Reviewer that helped us to improve the quality of the manuscript. We carefully responded to all the points and modified the manuscript accordingly.
Thanks and best regards.
Reviewer 1
Comments and Suggestions for Authors
The manuscript needs some work before it can be published. Some missing data should be presented in the manuscript. Below are some questions/ comments for the authors to improve the manuscript.
We are grateful for the comments and suggestions by the Reviewer that helped us to improve the quality of the manuscript. We carefully responded to all the points and modified the manuscript accordingly.
Comments
- Citation section in the first page for “polymers” journal not for “coating”, please revise.
Response: The reviewer's comment is much appreciated. We changed the text accordingly.
- The abstract should be concerned with actual results rather than posted information,
Response: We thank the reviewer for the suggestion. We changed the abstract accordingly.
- The abbreviation should be mentioned completely the first time, please check throughout the manuscript.
Response: We thank the reviewer for the notice. We updated the text and included an abbreviation list at the end of the manuscript as well.
- Keywords should contain words about the main hypothesis of the study, the author mentioned "Nanoparticles; Nanomaterials" it is the same meaning; " Antimicrobial; Antibacterial" also the same meaning?
Response: The reviewer's notice is greatly appreciated. However, because we know that researchers use a range of phrases in search engines, we attempt to incorporate all the terminology to ensure that our study is findable.
- The introduction contains huge information not related together, and in most cases, the information is repeated. Also, the introduction contains more sub-titles, and this is not preferred in the introduction. therefore, authors should rephrase the introduction.
Response: The reviewer's comment regarding this issue is greatly appreciated and very helpful! We revised the introduction accordingly.
- The authors should be adding a clear hypothesis at the end of the introduction.
Response: The reviewer's advice is much appreciated, your suggestions have been taken into consideration. We made the necessary changes to the introduction.
- Line 74, “(Fe3O4)” should be corrected to “Fe3O4”.
Response: The reviewer's advice is much appreciated. We changed the text accordingly.
8- Line 86, “gram positive and gram negative” should be corrected to “Gram-positive and Gram-negative”.
Response: The reviewer's advice is much appreciated. We changed the text accordingly.
9- Figure 1 repeated twice, Also, authors should be adding a clear title for figures containing the definition for all abbreviations in figures.
Response: a) The reviewer's comment on this subject is appreciated and useful! These errors were not displayed in the word document! We deleted the cross-links between figures captions and text to avoid such error appearing during the production process. Thank you for bringing it to our attention, we were able to resolve the issue. b) We appreciate the reviewer’s suggestion. We added the text accordingly.
10- At the “2.2. Data collection” section, authors said that “A literature search was conducted on articles published between 2010 and 2021 that studied…..”, by simple online search found some published studies not included in the survey such as :https://doi.org/10.1016/j.carbpol.2012.08.064, http://doi:10.3390/antibiotics9100641,https://doi.org/10.1021/acs.iecr.0c04880, Please check.
Response: We appreciate the reviewer’s comment. However, we did not want to imply that we considered all research from 2010 to 2021, rather, we intended to indicate that we selected certain studies based on specific keywords (antimicrobial, antibacterial, nanomaterial, nanoparticles, washes, textile, bactericidal) between 2010 and 2021. We changed the text accordingly.
https://www.mdpi.com/2079-6382/9/10/641/htm: this study reports the antibacterial capacity expressed as MIC. This study would not fit the structure of the outcome that we predict (%).
https://doi.org/10.1016/j.carbpol.2012.08.064, https://pubs.acs.org/doi/full/10.1021/acs.iecr.0c04880 : those studies could definitely be included as it fits the methodology and the purpose of the manuscript. The fact hat we missed this article, clearly suggests that a manual data extraction process is highly time consuming and inappropriate. Again, this stress the fact that a source of combined information is required, in order for researchers not to miss information. We appreciate the reviewers’ comments, however, including new data at this stage will require additional time such as updating the dataset, cleansing the data, re-training the models and changing all the findings, in our study. We believe, that for the demonstration purposes, in this first attempt to create such a model, the data are sufficient. We will incorporate reviewers’ studies in our future work.
11- Materials and Methods contain sentences whose correct place is introduction such as lines 167-172.
Response: The reviewer's advice is much appreciated. We changed the text accordingly.
12- Line 279, “gram-positive, gram-negative and fungi” should be “Gram-positive, Gram-negative, and fungi”.
Response: The reviewer's comment is much appreciated. We changed the text accordingly.
13- Line 282, “…test)” delete “)”.
Response: The reviewer's comment is much appreciated. We changed the text accordingly.
14- Figure 2b, please add the complete scientific name of organisms in figure footnotes.
Response: The reviewer's advice is much appreciated and thank you for bringing this to our attention. We changed the text in the table and figure 2b accordingly.
15- Please check the figure numbers throughout the manuscript in the figure title and text.
Response: The reviewer's comment is much appreciated. We changed the text accordingly.
16- Figure 2, repeated twice on pages 10 and 11, Also, the number of this figure is incorrect.
Response: The reviewer's comment on this subject is appreciated. These errors were not displayed in the word document. We fixed the caption figures error as indicated in a previous comment. Thank you for bringing it to our attention, we were able to resolve the issue.
17- What is the meaning of “Error! Reference source not found” it is repeated more than one time
Response: The reviewer's comment on this subject is appreciated and useful! These errors were not displayed in the word document! We fixed the caption figures error as indicated in a previous comment. Thank you for bringing it to our attention, we were able to resolve the issue.
- The manuscript contains numerous sentences that are not clear and require rephrasing.
Response: The reviewer's remark regarding this issue is greatly appreciated and very helpful. We revised the manuscript accordingly.
19- In conclusion, the authors should explain how researchers benefit from this model and its applicability?
Response: The reviewer's advice is much appreciated and thank you for bringing this to our attention. We changed the text accordingly.
20- The manuscript contains a high number of references “135 references”, I recommended reducing it.
Response: The reviewer's comment is appreciated. We reduced the references to 121.
21- The manuscript should be subjected to deep English editing.
Response: The reviewer's remark regarding this issue is greatly appreciated and very helpful! We revised the manuscript accordingly.
Reviewer 2 Report
I've done my review of the manuscript entitled "A supervised machine-learning prediction of textile’s antimicrobial capacity coated with nanomaterials. by Mahsa Mirzaei and colleagues.
The authors sought to create a machine learning technique to predict the antibacterial efficiency of nanotextiles in this paper. They investigated several regression algorithms and discovered that the RF model delivers superior prediction with 70% accuracy. The attribute significance denotes the relevance of the p-chem characteristics of NM and the application/deposition technique in predicting the antibacterial efficacy of nanotextile.
I find the topic very interesting and there are many advantages with this study carried out by the authoring team and the result is sufficient for publication in a good academic journal like Coating. I suggest minor corrections.
Suggestion: Discussion is not clearly presented and is flawed and frustrating seriously. This part should substantiate with supportive work done previously. I would like to see the revised version.
Author Response
We are grateful for the comments and suggestions by the Reviewer that helped us to improve the quality of the manuscript. We carefully responded to all the points and modified the manuscript accordingly.
Thanks and best regards.
Reviewer 2
Comments and Suggestions for Authors
I've done my review of the manuscript entitled "A supervised machine-learning prediction of textile’s antimicrobial capacity coated with nanomaterials. by Mahsa Mirzaei and colleagues.
The authors sought to create a machine learning technique to predict the antibacterial efficiency of nanotextiles in this paper. They investigated several regression algorithms and discovered that the RF model delivers superior prediction with 70% accuracy. The attribute significance denotes the relevance of the p-chem characteristics of NM and the application/deposition technique in predicting the antibacterial efficacy of nanotextile.
I find the topic very interesting and there are many advantages with this study carried out by the authoring team and the result is sufficient for publication in a good academic journal like Coating. I suggest minor corrections.
Suggestion: Discussion is not clearly presented and is flawed and frustrating seriously. This part should substantiate with supportive work done previously. I would like to see the revised version.
We are grateful for the comments and suggestions by the Reviewer that helped us to improve the quality of the manuscript. We carefully responded to all the points and modified the manuscript accordingly.
Response: The reviewer's advice is greatly appreciated. We changed the discussion section accordingly.
Reviewer 3 Report
Abstract
Write in full `p-chem properties` as in current state it is confusing or abbreviate the physicochemical properties as first when it appears, then used subsequently as p-chem.
Authors missed out some latest, seminal report on the topic to cite. For example, (2) disruption of the microbial cell membrane [41, 42], please cite a work https://doi.org/10.1371/journal.pone.0175428 showing how Nano whisker physical parameters and chemistry demonstrate membrane damage based on pining of bacteria on nanostructures.
Write all the in vitro terms in italics.
Line 278 and 281, I see typos (Error! Reference source not found.b). If author deliberately put these captions while implementing and validating the model, I suggest removing the error statement as it looks clumsy in scientific paradigm.
Figurer 1 duplicated on page 4 and 5. I suggest removing it from page 5 retaining at page 4 since figure captions give the different segment of the figures in details, which is useful for the readers.
I suggest changing type of washing machine (durability test) to durability test to make it more scientific.
In section 2.2. Data collection, please how many papers were collected for each specific keywords and were the search limited to title/abstract or whole papers authors found in search engines
In figure 5, change x-axis labels for nanomaterials `NMS` to `NMs`. Change Concetration to Concentration (typo).
Statements regarding regression models, cite https://doi.org/10.1002/aisy.202000084 a latest report explaining the Random forest (RF) applicability in context with data preprocessing along line 220-224 to make the reference list up to date.
Please check carefully typos in designating materials (e.g. Fe3o4, TiO2, line 307-308 type of biner etc.) or other modelling parameters throughout the draft.
Some abbreviations noted which are not presented in full form, which hinders the flow of draft reading and understanding. Add a separate segment with all abbreviations used in the study.
I see figure 2 twice with the cations, please correct those anomaly carefully checking the draft from beginning and making sure figures are cited appropriately in the text.
Author Response
We are grateful for the comments and suggestions by the Reviewer that helped us to improve the quality of the manuscript. We carefully responded to all the points and modified the manuscript accordingly.
Reviewer 3
We are grateful for the comments and suggestions by the Reviewer that helped us to improve the quality of the manuscript. We carefully responded to all the points and modified the manuscript accordingly.
Comments and Suggestions for Authors
- Write in full `p-chem properties` as in current state it is confusing or abbreviate the physicochemical properties as first when it appears, then used subsequently as p-chem.
Response: The reviewer's advice is greatly appreciated and thank you for bringing this to our attention. We changed the text accordingly. Line 24
- Authors missed out some latest, seminal report on the topic to cite. For example, (2) disruption of the microbial cell membrane [41, 42], please cite a work https://doi.org/10.1371/journal.pone.0175428 showing how Nano whisker physical parameters and chemistry demonstrate membrane damage based on pining of bacteria on nanostructures.
Response: The reviewer's suggestion is appreciated, and we added the citation to the text accordingly. Line 93.
- Write all the in vitro terms in italics.
Response: The reviewer's comment is much appreciated. We changed the text accordingly.
- Line 278 and 281, I see typos (Error! Reference source not found.b). If author deliberately put these captions while implementing and validating the model, I suggest removing the error statement as it looks clumsy in scientific paradigm.
Response: The reviewer's remark regarding this issue is greatly appreciated and very helpful! These mistakes did not show up in the word document! We removes the figures captions that are crosslinked with the text to avoid those mistakes during production. Thank you for bringing it to our attention, we were able to resolve the issue.
- Figurer 1 duplicated on page 4 and 5. I suggest removing it from page 5 retaining at page 4 since figure captions give the different segment of the figures in details, which is useful for the readers.
Response: The reviewer's remark regarding this issue is greatly appreciated and very helpful! These mistakes did not show up in the word document! Thank you for bringing it to our attention, we were able to resolve the issue.
- I suggest changing type of washing machine (durability test) to durability test to make it more scientific.
Response: The reviewer's suggestion is greatly appreciated. We changed the text accordingly.
- In section 2.2. Data collection, please how many papers were collected for each specific keywords and were the search limited to title/abstract or whole papers authors found in search engines.
Response: The reviewer's comment is appreciated. We searched for papers containing the keywords Nanoparticles, Nanomaterial, Antibacterial, Antimicrobial, Textile, bactericidal, fabric with and without washes in the title. We chose papers that included all these characteristics (nanoparticles, antimicrobial assessment, and textiles). b) The search required checking the entire manuscript to see what methodology they used to investigate the antibacterial properties.
Antibacterial 37, Antimicrobial 22, washes 1, cycles 1, nanoparticles 47, textile 23, Fabric 40, Fiber 3.
- In figure 5, change x-axis labels for nanomaterials `NMS` to `NMs`. Change Concetration to Concentration (typo).
Response: We appreciate the reviewer’s notice. We changed the text accordingly. Line 356.
- Statements regarding regression models, cite https://doi.org/10.1002/aisy.202000084 a latest report explaining the Random forest (RF) applicability in context with data preprocessing along line 220-224 to make the reference list up to date.
Response: The reviewer's advice is much appreciated. We added the citation accordingly.
- Please check carefully typos in designating materials (e.g. Fe3o4, TiO2, line 307-308 type of biner etc.) or other modelling parameters throughout the draft.
Response: The reviewer's advice is much appreciated. We changed the text accordingly. Line 83,209,371.
- Some abbreviations noted which are not presented in full form, which hinders the flow of draft reading and understanding. Add a separate segment with all abbreviations used in the study.
Response: The reviewer's comment is greatly appreciated. We updated the text and included an abbreviation list.
- I see figure 2 twice with the cations, please correct those anomaly carefully checking the draft from beginning and making sure figures are cited appropriately in the text.
Response: The reviewer's remark regarding this issue is greatly appreciated and very helpful! These mistakes did not show up in the word document! Thank you for bringing it to our attention, we were able to resolve the issue.
Thanks and best regards.
Reviewer 4 Report
The manuscript describes, “A supervised machine-learning prediction of textile’s antimicrobial capacity coated with nanomaterials.” which is suitable for Journal of Coating. Anyhow, the reviewer would like to make the following comments
- I recommend that add more application of nanomaterials in the line 59, page 2.
- Justify why the Mirzaei’s tool can be an effective model to predict the antimicrobial properties of the nanomaterials.
- What is the novelty in this study?
- What do you mean “Inclusion requirements consist of: English language”
- Write the full name of “ Cs” in the line 168
- Authors listed the some of the non-toxicity of the nanomaterials in the line 168. However, recently nano-TiO2 shows an unsuitable behavior (org/10.3390/nano11092354). Consequently, NMs need to more research to reveal the intrinsic behavior on the human. So, justification is recommended.
Author Response
We are grateful for the comments and suggestions by the Reviewer that helped us to improve the quality of the manuscript. We carefully responded to all the points and modified the manuscript accordingly.
Reviewer 4
Comments and Suggestions for Authors
The manuscript describes, “A supervised machine-learning prediction of textile’s antimicrobial capacity coated with nanomaterials.” which is suitable for Journal of Coating. Anyhow, the reviewer would like to make the following comments
We are grateful for the comments and suggestions by the Reviewer that helped us to improve the quality of the manuscript. We carefully responded to all the points and modified the manuscript accordingly.
- I recommend that add more application of nanomaterials in the line 59, page 2.
Response: We thank the reviewer for the suggestion. We changed the text accordingly.
- Justify why the Mirzaei’s tool can be an effective model to predict the antimicrobial properties of the nanomaterials.
Response: The reviewer's advice is much appreciated and thank you for bringing this to our attention. We updated the text accordingly.
This paper introduces a novel usage of ML towards the prediction of antimicrobial capacities of nano-textiles by considering diverse deposition techniques. We bridge various p-chem properties, experimental conditions and testing approaches to predict the antimicrobial capacity after durability testing. Additionally, we demonstrate the attributes importance in predicting the textiles capacity to retain high antimicrobial capacity after washing cycles.
- What is the novelty in this study?
Response: The reviewer's advice is much appreciated and thank you for bringing this to our attention. We updated the text accordingly.
This approach can assist scientist predict the characteristics that impact the antimicrobial capacities of the nano-textile, allowing them to improve nano-textiles and therefore minimizing the growth of harmful microorganisms. The ability to forecast antimicrobial properties of nano-textiles, particularly in health care settings, offers enormous potential in terms of providing tangible health care benefits.
- What do you mean “Inclusion requirements consist of: English language”
Response: The reviewer's feedback is appreciated. This refers to all research that has been published in English (not in Greek, Persian, etc)
- Write the full name of “ Cs” in the line 168
Response: The reviewer's feedback is appreciated. In the introduction, we give the full name of Cs, lane 76 (chitosan (Ch) [38] or caesium (Cs) [39]).
- Authors listed the some of the non-toxicity of the nanomaterials in the line 168. However, recently nano-TiO2 shows an unsuitable behavior (org/10.3390/nano11092354). Consequently, NMs need to more research to reveal the intrinsic behavior on the human. So, justification is recommended.
Response: The reviewer's advice is much appreciated. However, in our paper We that we gathered information on a number of NMs (Ag, Fe3O4, TiO2, SiO2, MgO, CuO, ZnO, ZrO2, Au, Ch, Cs) used in textiles and discovered that these NMs are biocompatible.
org/10.3390/nano11092354, this study indicates that TiO2 has no unfavorable effects on human skin, since it cannot penetrate the deeper (inner) layers of human skin. As a result, textiles coated with TiO2 are biocompatible (no cell membrane damage has been documented) and have remarkable functional properties.
However, there is dispute on this topic, as evidenced by the following studies:
This study (https://doi.org/10.1016/j.impact.2020.100224) shows that ingested TiO2 can cause gut microbiota disorders as a result of toxicity caused by oral exposure to TiO2 NPs and highlight the need of additional research in this area.
Other research (doi: 10.2147/IJN.S249441) has found that Ti02 has minimal toxicity and good biocompatibility.
Therefore, we acknowledge the need for addition research and study on Ti02 and its safety in food, as well as its interactions with other compounds, is required in the future.
Thanks and best regards.
Round 2
Reviewer 1 Report
I would like to thank the authors for correcting most of the issues, but there are some issues that must be correct after accepting publications
1- The scientific name must be italic (especially in Table 1)
2- Line 303, please delete "Error! Reference source not found"
Reviewer 2 Report
Great work. Ok for publication in Coating
Reviewer 3 Report
accept